# Comparison of the Effects of *Yucca saponin*, *Yucca schidigera*, and *Quillaja saponaria* on Growth Performance, Immunity, Antioxidant Capability, and Intestinal Flora in Broilers

**DOI:** 10.3390/ani13091447

**Published:** 2023-04-24

**Authors:** Zhenglie Dai, Huixian Wang, Jinsong Liu, Haoran Zhang, Qing Li, Xiaorong Yu, Ruiqiang Zhang, Caimei Yang

**Affiliations:** 1College of Animal Science and Technology & College of Veterinary Medicine, Zhejiang Agricultural and Forestry University, Hangzhou 311300, China; 2Key Agricultural Research Institute of Veagmax Green Animal Health Products of Zhejiang Province, Anji 313300, China

**Keywords:** *Yucca saponin*, *Quillaja saponaria*, yellow-feather broilers, growth performance, intestinal flora

## Abstract

**Simple Summary:**

Antibiotics are banned in animal feed, and substances used to replace antibiotics are being found continuously. *Yucca schidigera*, *Quillaja saponaria*, and *Yucca saponin* are natural plants and their components, which have many beneficial health functions for animals, are suitable as animal feed additives to replace antibiotics. By comparing the impacts of three substances on the growth, immunity, and intestinal health of broilers, this study found that *Yucca schidigera*, *Quillaja saponaria*, and *Yucca saponin* all promote the healthy growth of broilers, and *Yucca saponin* has better effects than the other two substances.

**Abstract:**

The purpose of this study is to investigate the effects of *Yucca saponin* (YSa), *Yucca schidigera* (YS), and *Quillaja Saponaria* (QS) on growth performance, nitrogen metabolism, immune ability, antioxidant capability, and intestinal flora of yellow-feather broilers. This study randomly divided a total of 480 1-day yellow-feather broilers into 4 treatment groups. Factors in the 4 groups included CON group (basic diet), YSa group (basic diet mixed with 500 mg/kg YSa), YS group (basic diet mixed with 500 mg/kg YS), and QS group (basic diet mixed with 500 mg/kg QS). Throughout the 56-day study period, YSa, YS, and QS groups had higher average daily gain in broilers than the CON group (*p* < 0.01). The YS group had a lower feed gain ratio (F: G) in broilers than the CON group (*p* < 0.05). YSa, YS, and QS showed increased serum immunoglobin A (IgA), immunoglobin Y (IgY), immunoglobin M (IgM), and total antioxidant capacity (T-AOC) levels; enhanced acetic acid, butyric acid, and valeric acid levels of cecal content; and reduced contents of ammonia nitrogen, urea nitrogen, interleukin-6 (IL-6), tumor necrosis factor-α (TNF-α), and malondialdehyde (MDA) in serum in broilers (*p* < 0.05). The relative abundance of Lachnoclostridium in the QS group was decreased compared with that in the CON group (*p* < 0.05). Higher IgA and IgY sera contents were observed in the YS group compared to the YSa and QS groups (*p* < 0.05). In contrast with the QS group, the serum IL-6 concentration of the YS group was reduced (*p* < 0.05). In conclusion, YSa, YS, and QS promoted growth performance, nitrogen metabolism, immunity, antioxidant capability, and intestinal flora in broilers. Through the comparison of YSa, YS, and QS, it was found that YS is more suitable as a feed additive to ameliorate the healthy growth of broilers.

## 1. Introduction

Many important issues influence the health and well-being of broilers in commercial poultry operations. Unlike those of ruminants, poultry diets contain more nutrients. This leads to high nutrient wasting [1]. High protein in the diet may cause disorder of ammonia metabolism and ammonia emission [2]. Studies have shown that living in a high ammonia environment can lead to oxidative stress in poultry [3]. Oxidative stress can cause cell senescence of poultry [4], as well as gastrointestinal disorders, which may involve poor intestinal health and loss of production [5]. In order to alleviate oxidative stress and enhance the immunity of broilers, antibiotics are used in poultry production [6]. Antibiotic resistance has gradually spread among various animal pathogens and become a major issue of global public health [7]. Thus, the development of safe and green additives that can improve poultry growth performance and anti-oxidative stress ability, and ameliorate the metabolism of ammonia, has become one of the urgent problems to be solved in the poultry industry [8].

*Yucca schidigera* (YS) is an herbaceous plant in Liliaceae [9], which has anti-inflammatory, antioxidant, and immunostimulatory effects [10] and can significantly reduce ammonia emission [11]. *Yucca saponin* (YSa) is a natural active ingredient in YS, which is used as an additive in food [12]. Studies have shown that YSa can reduce ammonia in rumen and methane [13]. In an experiment involving the intragastric administration of *Eimeria oocyst* to broilers, broilers supplemented with YSa in their diet had higher anti-stress ability, immune function, and growth performance [14]. *Quillaja saponaria* (QS) contains a high concentration of saponin and belongs to the *Quillajaceae* family [15]. A previous study demonstrated that QS can enhance the immune ability of mice [16]. A study by Bartoš et al. [17] indicated that the diet of growing–finishing pigs supplemented with QS could reduce ammonia emission. Tipple et al. [18] reported that *Quillaja saponin* extract has good antioxidant properties.

Currently, there are few studies that compare the effects of YSa, YS, and QS on animal health. The aim of this experiment was to investigate the effects of YSa, YS, and QS on growth performance, ammonia metabolism, immunity, anti-oxidative capacity, and intestinal flora in broilers, and then judge their respective advantages. The present experiment can provide a theoretical reference for the further application of YSa, YS, and QS in the poultry breeding industry.

## 2. Materials and Methods

### 2.1. Animals, Diets, and Treatment

A total of 480 1-day male yellow-feather broilers (body weight, 32.00 ± 2.00 g) were stochastically divided into 4 treatment groups, with 8 replicates in each treatment group and 15 birds in each replicate. The chickens were raised in cages that were 2 m long, 1.5 m wide, and 2 m high. When the broiler weight reached 700 ± 50 g, seven broilers were separated and raised in adjacent cages. The test period was 56 days. Incandescent lamps were provided for the first 7 days. The lighting schedule was 24 h a day for the first 7 days of age, after which 23 h a day of light were provided until the end of the experimental period.

YSa, YS, and QS were obtained from Key Agricultural Research Institute of Veagmax Green Animal Health Products of Zhejiang Province. The YS used in this study was powder from the crushing of the whole plant. The YSa used in this study contained ≥30% saponin and a B50 value ≤ 4 mg, and was in powder form. The QS used in this study was 100% natural powder from whole-plant crushing. The addition amount of the three additives was 500 mg/kg according to previous research results [19]. The experiment included a control group (CON), YSa group, YS group, and QS group. The groups were fed with a basic diet, basic diet + 500 mg/kg YSa, basic diet + 500 mg/kg YS, and basic diet + 500 mg/kg QS, respectively. No antibiotic additives were included in the basic diet of each group.

The basic diet was formulated in line with the Nutritional Requirements for Yellow-Feather Broilers in China (NY/T 3645-2020) and the recommendations of China Broiler Feed Standard (NY/T 332004). The composition and nutritional level of the basic diet are listed in Table 1.

### 2.2. Sample Collection

One broiler of around average body weight from each repetition was selected on the 28th and 56th days. The broilers were dissected after taking 4 mL blood samples from the carotid artery. Blood was centrifuged at 4 °C, 4000× *g* for 10 min to separate serum. The serum was sub-packed in a 1.5 mL centrifuge tube and stored at −80 °C for subsequent experiments. After dissection and gently washing with sterilized normal saline, the jejunum segment (about 1 cm) was removed from the middle part of the jejunum. The jejunum was fixed with 4% formaldehyde solution (Aladdin, Shanghai, China) and preserved at normal atmospheric temperature for morphological evaluation. At the same time, the cecal contents were stored at −80 °C for analysis after extrusion into a sterile tube.

### 2.3. Growth Performance Calculation

Average daily gain, average daily feed intake and ratio of Feed: Gain are calculated according to the following formulas:$$\mathrm{Average}\;\mathrm{daily}\;\mathrm{gain}=\frac{\mathrm{Final}\;\mathrm{weight}\;-\;\mathrm{Initial}\;\mathrm{weight}}{\mathrm{Days}\;\times \;\mathrm{Number}\;\mathrm{of}\;\mathrm{broilers}}$$
$$\mathrm{Average}\;\mathrm{daily}\;\mathrm{feed}\;\mathrm{intake}=\frac{\mathrm{Final}\;\mathrm{feed}\;\mathrm{weight}\;-\;\mathrm{Initial}\;\mathrm{feed}\;\mathrm{weight}}{\mathrm{Days}\;\times \;\mathrm{Number}\;\mathrm{of}\;\mathrm{broilers}}\;$$
$$\mathrm{Feed}:\;\mathrm{Gain}=\frac{\mathrm{Average}\;\mathrm{daily}\;\mathrm{feed}\;\mathrm{intake}}{\mathrm{Average}\;\mathrm{daily}\;\mathrm{gain}}$$

### 2.4. Serum Related Indexes

The nitrogen-related indicators were: urea nitrogen, uric acid, xanthine oxidase (XOD), immunoglobulins; cytokine-related indicators were: immunoglobin A, immunoglobin M, immunoglobin Y, interleukin-1β (IL-1β), interleukin-6 (IL-6), and tumor necrosis factor-α (TNF-α). A commercial kit of Nanjing Jiancheng Institute of Biological Engineering was used to determine the antioxidant-related indicators: malondialdehyde (MDA), total antioxidant capacity (T-AOC), and glutathione peroxidase (GSH-PX). Refer to the kit manual for the specific operation process.

### 2.5. Fecal Nitrogen Related Indexes

The feces of broilers were collected from 8:00 to 9:00 on the 28th and 56th days of the experiment. A sterilized paper pad was laid under the cage, collected into a transparent plastic zip lock bag with tweezers and mixed well, and then frozen at −20 °C. During the test, 8 g of chicken feces was mixed with 24 mL of normal saline in a 50 mL centrifuge tube, put in a water bath pot, and bathed in water at 50 °C for half an hour. After the water bath, the fecal turbid liquid was placed on a shaking table and shaken at 37 °C for 180 r for 30 min. After removing from the shaking table, it was centrifuged at 6000× *g* for 10 min under normal temperature. A quantity of 1 mL of supernatant was extracted into a 1.5 mL centrifuge tube and centrifuged at 6000× *g* for 5 min under normal temperature. From the supernatant, the contents of uric acid, uricase, urea nitrogen, and ammonia nitrogen in feces were determined using the commercial kit (Nanjing Jiancheng Bioengineering Institute, Nanjing, China).

### 2.6. Jejunal Morphology Analysis

The 0.5 cm jejunum segment was fixed with 4% formaldehyde solution for 48 h, embed in paraffin (4 µm) after dehydration, and dyed with hematoxylin-eosin. The slices were viewed under a fluorescence microscope (DS-FI2 camera, EclipseCi, Nikon, Japan). In 10 visual fields of each sample, the villus height and crypt depth were measured at the magnification of 40 times. The villus height was calculated from the top of villus to the junction of the villus and crypt. The crypt depth was defined as the depth between two adjacent villi. Based on these measurements, the ratio of villus height to crypt depth (VCR) was expressed by dividing the villus height by crypt depth.

### 2.7. Short-Chain Fatty Acid Determination

About 0.5 g cecal content was loaded into a 2 mL centrifuge tube, precooled ddH_2_O was added according to the mass volume ratio of 1:3, the sampling amount and the proportion of water were recorded, the centrifuge tube was placed on a vortex oscillator, the contents were thawed in the tube and mixed evenly, and rotated at 12,000× *g* at 4 °C; the supernatant was sucked after centrifugation for 10 min and 25% metaphosphoric acid was added according to a 5:1 volume ratio. This was mixed and placed in an ice bath for 30 min, before 12,000× *g* centrifugation at 4 °C for 10 min. The supernatant was taken from a 1 mL syringe, filtered through an inorganic phase filter head, and transferred to an injection bottle suitable for Agilent testing. A gas chromatograph (GC7890, Agilent Technologies, Santa Clara, CA, USA) was used with a HPFFAP column (Beijing pumeng Technology Co., Ltd., Beijing, China). After the machine test, the short-chain fatty acid (SCFA) content of each sample could be directly read out on the machine.

### 2.8. Microflora Determination

The genomic DNA of cecal contents was extracted by the SDS method, and the sequence of the V3–V4 region of the 16S rRNA gene was analyzed. The genomic DNA was amplified by three-step PCR. The amplified products were tested by electrophoresis-equal concentration mixing-redetection. The products were recovered using a recovery kit and sequenced on the Miseq platform. The sequence analysis of the V3–V4 region of the 16 rRNA gene and bioinformatics analysis of microbiome data were conducted by Beijing Novogene Technology Co., Ltd., Beijing, China.

### 2.9. Data Statistics and Analysis

Prism 8.0 software (GraphPad software company, Boston, MA, USA) and SPSS 25.0 software (IBM company, New York, NY, USA) were used for statistical analysis. One-way ANOVA was used to analyze the effective data, and Duncan’s method for multiple comparisons was used. Operational Taxonomic Units (OTUs) analysis, Nonmetric Multidimensional Scaling (NMDS) analysis, dilution curve analysis of cecal microflora, and Principal Component Analysis were conducted on the official website of Majorbio (https://vip.majorbio.com/, accessed on 19 July 2021). *p* < 0.05 was considered a significant difference, and *p* < 0.01 was considered an extremely significant difference. In the analysis chart of flora relative abundance results, “*” is significantly different, and “**” is extremely significantly different.

## 3. Results

### 3.1. Growth Performance

The specific data are shown in Table 2. On the 28th and 56th days of the test, the body weight of broilers in YSa, YS, and QS groups was higher than that of CON (*p* < 0.05). On days 1–28, there were higher average daily gains in broilers in YSa, YS, and QS groups than those in CON (*p* < 0.05). The average daily gain of broilers in the QS group increased more than that in YSa and YS groups (*p* < 0.05). The average daily feed intake of broilers in YSa, YS, and QS groups presented higher values than that in the CON group. The average daily feed intake of broilers in the YSa and QS groups was higher than that in the YS group (*p* < 0.05).

At the age of 29–56 days, the F: G of broilers was reduced by adding YSa and YS to the diet compared with the CON and QS groups (*p* < 0.05).

At the age of 1–56 days, the average daily gain of broilers in YSa, YS, and QS groups was higher than that in the CON group (*p* < 0.01). The average daily feed intake of broilers in the QS group improved by a large amount compared with that in the CON group (*p* < 0.05). Broilers in the YS group had a lower F: G than those in the CON group (*p* < 0.05).

### 3.2. Nitrogen Metabolism Indexes in Serum

Compared with the control group, the serum ammonia level of broilers in the YS group showed a decreasing trend at 28 days (*p* < 0.05, Figure 1). The serum uric acid content and XOD activity of broilers in YSa, YS, and QS groups decreased (*p* < 0.05). The level of serum uric acid in YSa group appeared to be significantly lower than that in the QS group (*p* < 0.05).

In contrast with the control group, the serum ammonia level of broilers in YSa, YS, and QS groups decreased at 56 days (*p* < 0.05). The addition of YSa to the diet reduced the content of uric acid in serum (*p* < 0.05), and the addition of YSa and QS reduced the level of urea nitrogen in serum (*p* < 0.05). Broilers in YSa and YS groups appeared to have lower XOD activity than those in CON and QS groups (*p* < 0.05).

### 3.3. Nitrogen Metabolism Indexes in Faecas

A decreasing trend in the content of ammonia nitrogen in the feces of 28-day broilers of the YSa and YS groups can be observed in Figure 2 (*p* < 0.05), compared with the CON group. Furthermore, the content of ammonia nitrogen in the YSa group appears to be much lower than that in YS group (*p* < 0.05). The addition of YSa to the diet reduced the levels of uric acid and urea nitrogen in the feces of broilers compared with those in the CON, YS, and QS groups (*p* < 0.05), and decreased urease in the feces of broilers compared with CON and QS groups (*p* < 0.05). The content of urease in the feces of broilers of the YS group appeared to decrease compared with that in the CON group (*p* < 0.05).

On the 56th day of the experiment, the contents of ammonia nitrogen and urea nitrogen in the feces of YSa, YS, and QS groups presented lower values than those in the CON group (*p* < 0.05). Compared with that in CON and YS groups, the content of uric acid in broilers feces of the QS group decreased (*p* < 0.05). The activities of urease in feces were reduced in YSa, YS, and QS groups compared with those in the CON group (*p* < 0.05). YSa reduced the activity of urease compared with QS (*p* < 0.05).

### 3.4. Serum Immunoglobulin

An increasing trend was seen in the contents of IgA and IgM in sera of the 28-day broilers of YSa, YS, and QS groups, as shown in Figure 3, compared with those in the CON group (*p* < 0.05). YSa and QS increased the content of IgY in serum compared with that in the CON group (*p* < 0.05). The YSa group showed a higher IgY level in serum than that in the YS group (*p* < 0.05).

A higher content of IgY in the serum of 56-day broilers in the YS group was seen compared to that in CON, YSa, and QS groups (*p* < 0.05). Higher contents of IgA and IgM in sera of broilers in YSa and YS groups were seen compared to those in the CON group (*p* < 0.05). Compared with YS and QS groups, the YSa group showed an increased content of IgM in serum. Compared with YSa and QS groups, the YS group showed an increasing trend in the content of IgA in serum (*p* < 0.05).

### 3.5. Serum Cytokines

A decreasing trend in the contents of IL-6 and IL-1β in sera of the 28-day broilers of YSa, YS, and QS groups was seen, as shown in Figure 3, compared with those of the CON group (*p* < 0.05). The content of TNF-α in the serum of broilers in YSa and QS groups showed a reduction compared with that in the CON group (*p* < 0.05). Moreover, the YSa group presented a lower content of TNF-α in serum than the YS group (*p* < 0.05).

Compared with the CON group, the addition of YSa, YS, and QS in feed reduced the IL-6 and TNF- α in serum in 56-day broilers (*p* < 0.05). YSa and YS groups had an IL-6 level reduction compared with the QS group (*p* < 0.05). A decreasing level of IL-1β appeared in the serum of broilers in YSa and YS groups compared with the CON group (*p* < 0.05).

### 3.6. Serum Antioxidant Enzymes

The 28-day broilers in the YSa, YS, and QS groups showed higher levels of T-AOC and GSH-PX in serum than those in CON group (*p* < 0.05, Figure 4), while the value of MDA showed a decreasing trend compared with that in the CON group (*p* < 0.05).

Increasing levels of T-AOC were seen in the serum of 56-day broilers in the YSa, YS, and QS groups (*p* < 0.05), while the content of MDA decreased compared with that in the CON group (*p* < 0.05).

### 3.7. Jejunum Morphology

The morphology of jejunum of 28-day and 56-day broilers is shown in Figure 5. Compared with the CON group, 28-day broilers in YSa, YS, and QS groups had an increasing value of the villus height and the VCR of the jejunum, while the crypt depth of the jejunum of broilers in YS and QS groups decreased (*p* < 0.05). Compared with YSa and QS groups, the villus height of the YS group increased (*p* < 0.05). Furthermore, YS and QS groups showed a higher VCR than that in the YSa group (*p* < 0.05). YS decreased the crypt depth in contrast with that of YSa (*p* < 0.05).

In 56-day broilers, the villus height and VCR of the jejunum in YSa, YS, and QS groups showed higher values than those in the CON group (*p* < 0.05), while the crypt depth presented lower values than those in the CON group (*p* < 0.05). In contrast with YSa and QS groups, the villus height of the YS group increased (*p* < 0.05). A higher VCR of broilers was seen in the YS group than that in YSa and QS groups (*p* < 0.05). Compared with the YSa group, the crypt depth of broilers in YS and QS groups decreased (*p* < 0.05).

### 3.8. Short-Chain Fatty Acid

The results of SCFA detection are shown in Figure 6. At 28 d, increasing levels of acetic acid were seen in cecal contents of broilers in the YSa group compared with those of the CON group (*p* < 0.05). The addition of YSa, YS, and QS to the diet raised the levels of isobutyric acid and isovaleric acid in the cecal contents of broilers in contrast with those in the CON group (*p* < 0.05). YSa and YS groups presented higher values of the contents of acetic acid and butyric acid than those in the QS group (*p* < 0.05). The levels of propionic acid and valeric acid in the caecum contents of broilers in the YS group increased compared with those in the QS group (*p* < 0.05).

At 56 d, in contrast with those of the CON group, the YSa, YS, and QS groups showed increasing levels of acetic acid, butyric acid, and valeric acid in the cecal contents of broilers (*p* < 0.05). High levels of propionic acid in the cecal contents of broilers in the YS group were seen compared with those in the CON group (*p* < 0.05). Moreover, compared with those in the CON group, the levels of isobutyric acid in the cecal contents of broilers in YS and QS groups improved (*p* < 0.05).

### 3.9. Cecal Microbiota

The relative abundance and diversity of cecal microflora of broilers on the 56th d of the experiment are shown in Figure 7. The Venn diagram shows that the four groups have 1722 common OTUs, and the CON, YSa, YS, and QS groups have 198, 168, 258, and 274 unique OTUs, respectively (Figure 7A). The stress value of the NMDS model is 0.076. It can be seen from the figure that the cecal microflora of broilers had a significant difference between that in the YSa, YS, and QS groups, and that in the CON group (Figure 7B). The dilution curve shows that the curve trend of YS, YSa, and QS groups in the horizontal direction is relatively flat, indicating the rationality of the sample data volume. It indirectly shows that under the same data volume, the sample abundance of the QS group is higher than that of YS and YSa groups (Figure 7C). The results of PCA showed a dramatic difference in the composition of the cecal flora community of broilers between that in the YSa, YS, and QS groups, and that in the CON group (Figure 7D). Moreover, the composition of the cecal flora community of broilers in YSa and YS groups was relatively similar.

Bacteroidota, Firmicutes, Euryarchaeota, Verrucomicrobiota, and unidentified bacteria were the dominant bacteria in the cecal microbiota of broilers (Figure 8A). Among them, the relative abundance of *Bacteroidota* in broilers in the YS group showed a higher value than that in YSa and QS groups (*p* < 0.05, Figure 8D). In all samples, *Bacteroides*, *Methanobrevibacter*, *CHKCI001*, *Allstipes*, and *Akkermansia* were the dominant bacteria (Figure 8B). The relative abundance of *Lachnoclostridium* decreased in YSa, YS, and QS groups. Furthermore, the relative abundance of *Lachnoclostridium* significantly declined when QS was added to the feed (*p* < 0.05, Figure 8E). At the species level, *Bacteroides sp-arseille-p3166*, *Bacteroides-gallinaceum*, *Bacteroides-barnesiae*, *Bacteroides-caecicola*, and *Bacteroides-caecigalllinarum* were the dominant strains (Figure 8C). In contrast with the CON group, an increasing level of *Bacteroides-barnesiae* of broilers in YSa, YS, and QS groups was shown (*p* < 0.05). Decreasing levels of *Desulforibrio-piger* of broilers appeared in YSa, YS, and QS groups compared with those in the CON group (*p* < 0.05, Figure 8F).

## 4. Discussion

Numerous studies have shown that dietary QS supplementation increased the body weight, improved average daily gain, and ameliorated feed conversion efficiency [20,21]. Adding YSa to the broiler diet was beneficial to increasing the average body weight [22]. Mohamed et al. [23] reported that the final body weight of broilers increased when supplemented with YS. Results of this experiment also found that YSa, YS, and QS improved the weight and ameliorated feed conversion efficiency of broilers, which is consistent with previous experiments. Previous studies have suggested YS extract can regulate the intestinal microflora and improve the full digestion and absorption of nutrients [24], while increasing absorption of nutrients, mainly nitrogen [25]. Thus, YS extract enhances the animals’ growth. The addition of YS extract not only promoted the growth performance, but also raised the concentrations of serum T-AOC and IgY, and the content of serum ammonia declined [24]. Therefore, the improvement in the growth performance of broilers may have a bearing on the improvement in nitrogen metabolism, anti-inflammatory ability, immunity, antioxidant capability, and intestinal health caused by YSa, YS, and QS.

Nitrogen in poultry excreta exists in different forms, such as uric acid and ammonia [26]. In the process of fecal decomposition, microorganisms in feces can change the nitrogen-containing compounds of feces, and produce harmful gases such as ammonia [27]. Research showed that the change in the nitrogen source in the poultry diet can affect the content of urea and uric acid in serum, meaning that high-quality additives can reduce blood ammonia content [28]. In this study, we found YSa, YS, and QS decreased the concentrations of ammonia, uric acid, urea nitrogen, and XOD in serum, and decreased ammonia nitrogen, uric acid, urea nitrogen, and urease in the feces of broilers. YSa is beneficial in improving nitrogen metabolism. It was found that saponins, which are an important component in YSa, reduced the ammonia content in muscle by delaying the accumulation of serum urea nitrogen and lactic acid [29]. Scheuermann et al. [30] mentioned that the reduction in the ammonia in the intestine can affect the blood ammonia concentration, which is consistent with this study. It was found that YS in the broiler diets can significantly reduce the indoor ammonia concentration in poultry farms [20]. Fan et al. [31] found that adding YSa to the diet reduced the nitrogen emission of piglets. Research by Patra and Yu [32] showed that QS reduced the concentration of ammonia in an in vitro rumen model, which may be due to the combination of saponins with ammonia or the reduction in the rate of ammonia release from the microbial decomposition of organic matter [33]. Therefore, YSa, YS, and QS may improve blood and fecal nitrogen metabolism via combining saponins with ammonia.

Tumor necrosis factor is a common type of proinflammatory cytokine [34]. IL-6 plays the most important role in promoting endocrine metabolism [35]. In this study, the decreasing results of IL-6 and TNF-α revealed that YSa, YS, and QS can inhibit inflammation in broilers. YSa, YS, and QS may alleviate inflammation in different ways. YS is rich in resveratrol, which can cause substantial reductions in TNF-α and IL-1β, and increase anti-inflammatory cytokine IL-10 [36]. On the other hand, saponins, as the main active substance in YSa, YS, and QS, can reduce the inflammatory response in vivo [9]. The protopanaxadiol saponin fraction (PPD-SF) diminished the release of some inflammatory cytokines such as nitric oxide and TNF-α, blocking p38, c-Jun N-terminal kinase (JNK), and other proteins by downregulating mRNA gene expression [37]. This may be the reason that YSa, YS, and QS alleviate the inflammation in broilers. YS and QS can be used as immune regulators by stimulating, inhibiting, or regulating immune response [38]. QS saponins can improve the output of immunoglobulin, and induce further cellular and humoral immunity [39]. Research has found that the addition of YS can linearly increase the IgG content in the serum of laying hens [40]. In this study, higher contents of IgA and IgY were found in the serum of YS group broilers than those in the YSa and QS groups, indicating that YS may have more advantages for improving immunity of broilers. Furthermore, the effect of YSa and YS on reducing serum inflammatory factors is better than that of QS.

Excessive free radicals, such as the superoxide anion radical (O_2_^−^) and hydroxyl radical (OH^−^), produced in the normal physiological process of animals will cause damage to biological macromolecules, including sugar, protein, and nucleic acid, and eventually induce biological macromolecules to lose function or cause dysfunction of metabolism [41]. T-AOC reflects the total antioxidant capacity of various antioxidant molecules and enzymes. MDA, the product of lipid peroxidation, is a biomarker used to determine whether an organism is in a state of oxidative stress [42]. GSH-PX counteracts oxidative damage, first in response to the excess production of free radicals [43]. The YS supplementation decreased the serum MDA concentration and increased the serum GSH-PX activity of calves [44]. Adding QS extract to the basic diet of fish decreased the content of MDA in Nile tilapia (*Oreochromis niloticus*) [45]. The serum MDA contents of broilers in the YSa, YS, and QS groups in this experiment decreased, while the content of T-AOC and the serum GSH-PX activity of broilers in the YSa, YS, and QS groups increased. YSa, YS, and QS showed no significant differences in their effects on the antioxidant function of broilers. Saponin enhanced the Nrf2 signaling pathway, which is an antioxidant pathway [24], and increased GSH-PX [46]. Considering that all three substances contain saponin, this experiment suggested that the saponin in YSa, YS, and QS can raise the antioxidant level of broilers through modulating the Nrf2 signaling pathway.

Villus height, crypt depth, and the ratio of villus height to crypt depth are indicators commonly used to evaluate intestinal integrity, which can reveal some information about intestinal absorption [47]. Saponins have the function of ameliorating the intestinal absorption of nutrients [48]. Adding YSa could significantly increase the duodenal villus height and VCR of 42-day-old broilers, and also significantly increase the VCR of the jejunum [49]. The study of Bafundo et al. [26] showed that adding QS and YS can effectively reduce the crypt depth and improve the villus height of the ileum in broilers. The study mentioned that the intestinal effect caused by YS and QS is due to the change in the intestinal microbial composition or the mucosa itself, and both YS and QS are capable of bacterial cell lysis. Our results show that the reductions in harmful bacteria such as Lachnoclostridium and *Desulforibrio-piger* were conducive to the regulation of intestinal flora and had effects on intestinal morphology [26]. In this study, YS has more advantages for increasing villus height and improving VCR, whereas QS has more advantages for improving intestinal morphology.

SCFAs are the main final product of intestinal microorganism metabolism [50]. Their presence in the intestine indicates good intestinal health and microbial activity [51]. Threonine can be transformed into 2-oxbutyric acid and ammonia by threonine ammonia lyase encoded in Bacteroidota genomes, and then propionic acid is produced by fermentation and degradation of 2-oxbutyric acid [52]. An increase in Bacteroidota was detected in the YS group, which may be the reason why its propionic acid content was higher than that in the YSa and QS groups. In this experiment, the contents of acetic acid and butyric acid in the YSa, YS, and QS groups increased significantly. Research found a negative relationship between Lachnoclostridium and the contents of acetic acid and butyric acid [53]. Meanwhile, we found that Lachnoclostridium decreased in YSa, YS, and QS groups. Acetic acid and butyric acid can improve intestinal absorption [54]; this may be one of the reasons why YSa, YS, and QS enhanced the growth performance of broilers. Studies have shown an increasing value of total SCFA in cecal contents when adding YS to a dog’s diet, and acetic acid mainly increased in cecal contents [55], which was the same result as seen in this experiment. Results suggest that YSa, YS, and QS increased the content of SCFA in the intestine, which can not only promote growth performance, but also affect ammonia emission by reducing the fermentation intensity of animal protein feed [56]. SCFA regulates the function of intestinal epithelial cells through different mechanisms to improve the morphology of the intestine, and enhance the intestinal barrier and host metabolism [57]. These results reveal that YSa, YS, and QS may improve the growth performance by ameliorating intestinal morphology and the content of SCFA of broilers.

The intestinal flora structure is an important aspect that affects the growth performance [58]. In this experiment, YS and QS supplementation in the diet of broilers resulted in more unique OTUs than that of YSa. It was observed from β diversity analysis that the cecal flora in YSa, YS, and QS groups were significantly different from those in the CON group. Lachnoclostridium in the YSa, YS, and QS groups was significantly reduced, and Bacetroidota in the YS group were significantly increased. A correlation has been reported between Lachnoclostridium and the increase in TNF-α and IL-6 [59]. This indicated that YSa, YS, and QS may reduce the occurrence of inflammation by reducing the abundance of Lachnoclostridium in broilers. At the species level, a significantly increasing level of *Bacteroides-barnesiae* appeared and the level of *Desulforibrio-piger* decreased significantly in the three groups. *Desulforibrio* is a kind of sulfate-reducing bacteria. It causes damage to the intestine by decomposing sulfate in the intestine and producing hydrogen sulfide [60], indicating that the addition of YSa, YS, and QS improved the intestinal health by reducing harmful intestinal flora.

## 5. Conclusions

In conclusion, broilers receiving YSa, YS, and QS supplementation showed enhanced growth performance, antioxidant ability, anti-inflammatory ability, and immunity, and improved intestinal health and ammonia metabolism. Compared with YSa and QS, YS is more widely used as a feed additive. YS has more advantages for improving the feed gain ratio, immunity, and intestinal morphology, and reducing serum inflammatory factors of broilers.

## Figures and Tables

**Figure 1 animals-13-01447-f001:**
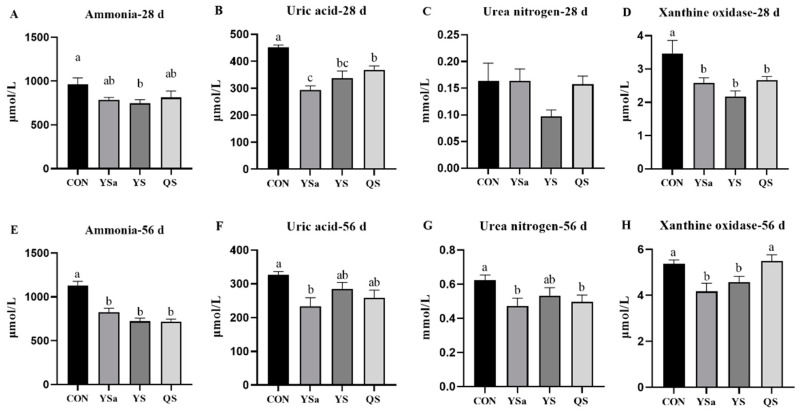
Effects of *Yucca saponin*, *Yucca schidigera*, and *Quillaja saponaria* on nitrogen metabolism indexes in serum of broilers. a–c Within a row, means with different superscripts differ significantly (*p* < 0.05).

**Figure 2 animals-13-01447-f002:**
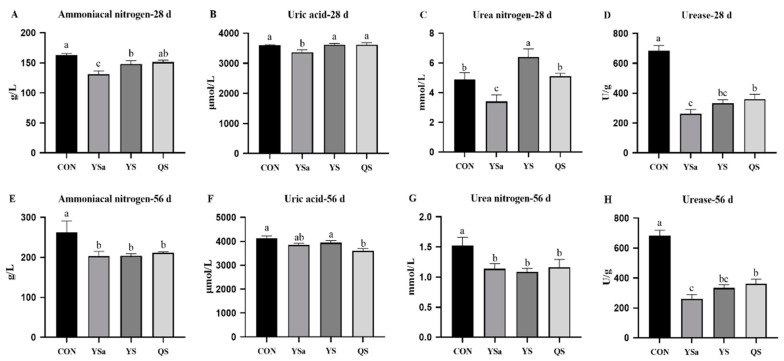
Effects of *Yucca saponin Yucca schidigera*, and *Quillaja saponaria* on nitrogen metabolism indexes in feces of broilers. a–c Within a row, means with different superscripts differ significantly (*p* < 0.05).

**Figure 3 animals-13-01447-f003:**
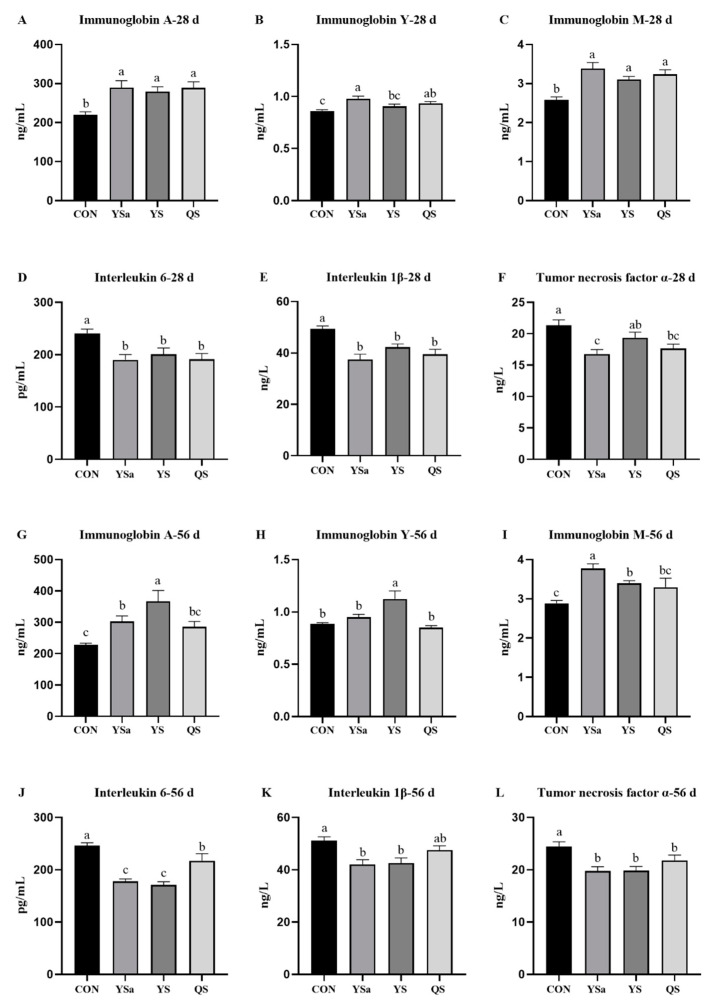
Effects of *Yucca saponin*, *Yucca schidigera*, and *Quillaja saponaria* on the concentrations of immunoglobulin and cytokines in serum of broilers. a–c Within a row, means with different superscripts differ significantly (*p* < 0.05).

**Figure 4 animals-13-01447-f004:**
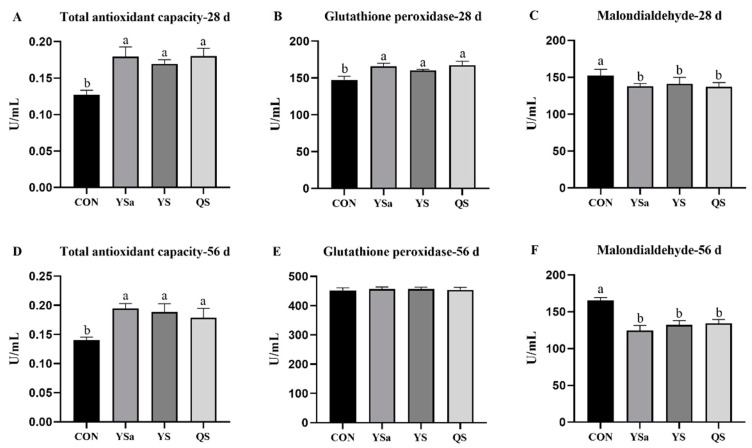
Effects of *Yucca saponin*, *Yucca schidigera*, and *Quillaja saponaria* on serum antioxidant activity of broilers. a, b Within a row, means with different superscripts differ significantly (*p* < 0.05).

**Figure 5 animals-13-01447-f005:**
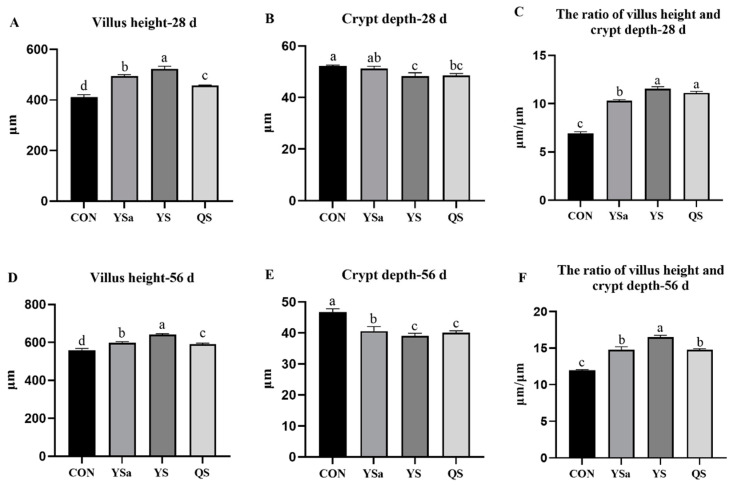
Effects of *Yucca saponin*, *Yucca schidigera*, and *Quillaja saponaria* on the morphology of the jejunum in broilers. a–d Within a row, means with different superscripts differ significantly (*p* < 0.05).

**Figure 6 animals-13-01447-f006:**
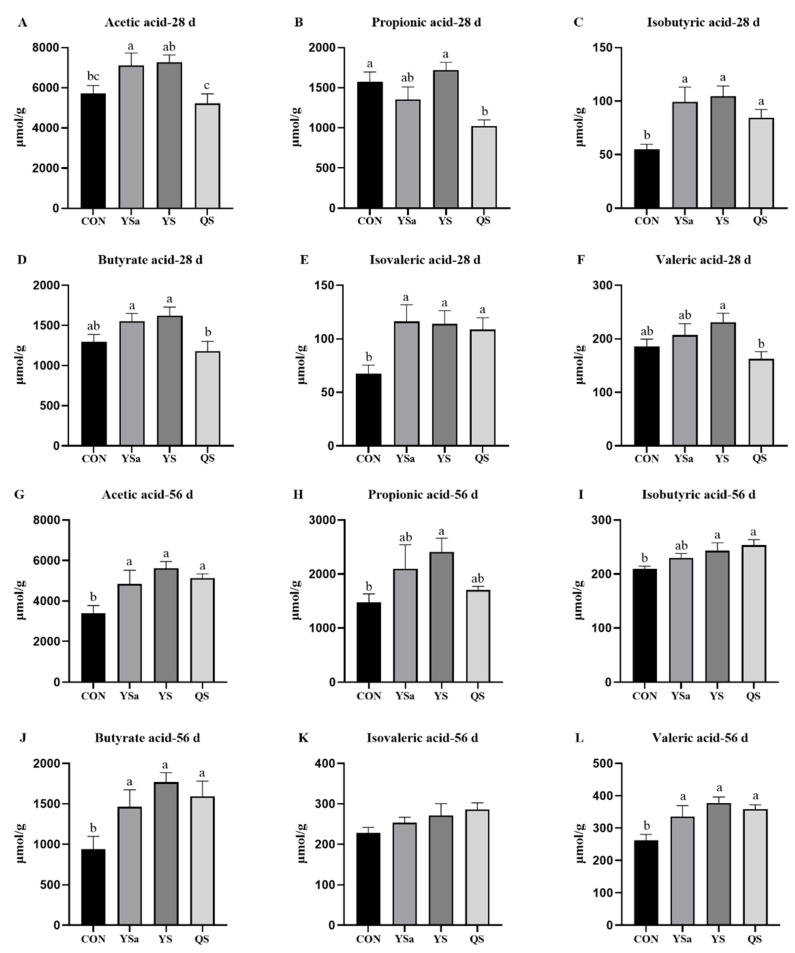
Effects of *Yucca saponin*, *Yucca schidigera*, and *Quillaja saponaria* on SCFA in 28-day old and 56-day old broilers. a–c Within a row, means with different superscripts differ significantly (*p* < 0.05).

**Figure 7 animals-13-01447-f007:**
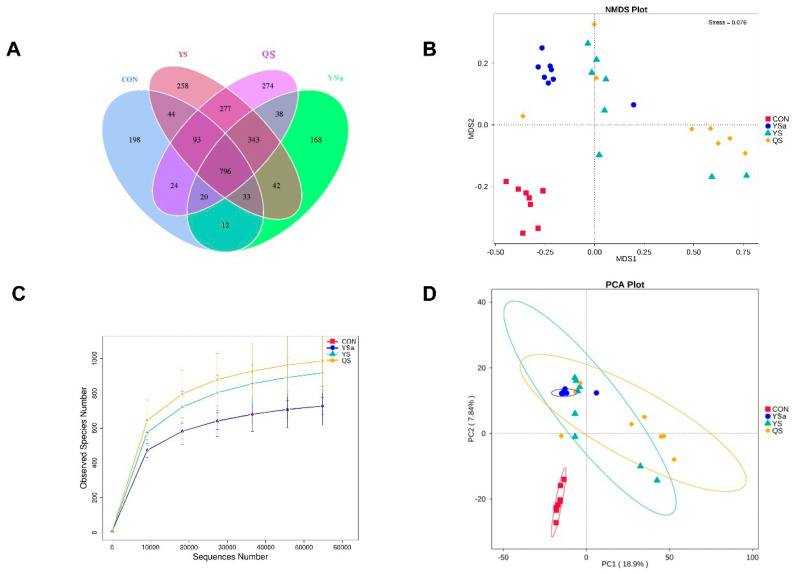
Effects of *Yucca saponin*, *Yucca schidigera*, and *Quillaja saponaria* on species clustering and complexity of cecal microbiota in 56-day old broilers ((**A**): Operational Taxonomic Units (OTUs) analysis of cecal microorganisms in 56-day broilers, (**B**): Nonmetric Multidimensional Scaling (NMDS) model of cecal microflora in 56-day broilers, (**C**): dilution curve of cecal microflora in 56-day broilers, (**D**): Principal Component Analysis of cecal microflora in 56-day broilers).

**Figure 8 animals-13-01447-f008:**
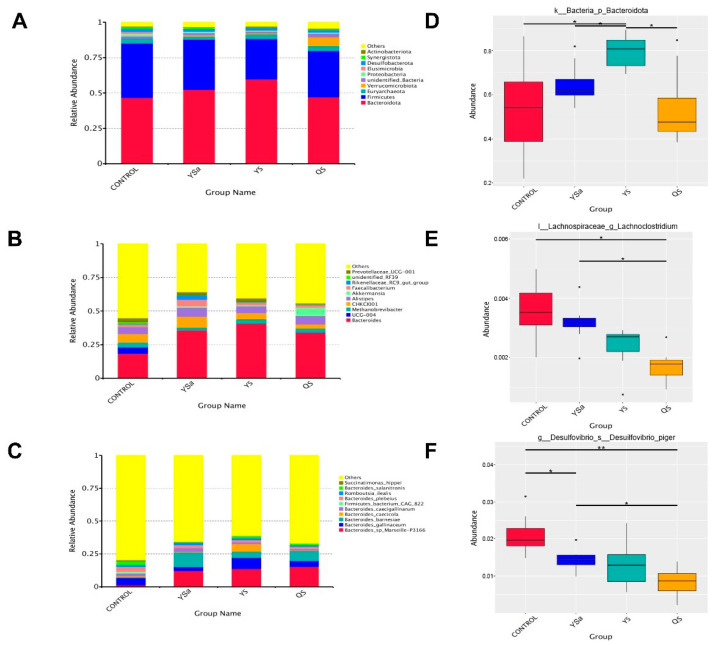
Effects of *Yucca saponin*, *Yucca schidigera*, and *Quillaja saponaria* on cecal microflora at phylum, genus, and species levels in 56-day-old broilers. ((**A**): Phylum-level species abundance, (**B**): genus-level species abundance, (**C**): species-level species abundance, (**D**): relative abundance of Bacetroidota in each group, (**E**): relative abundance of *Lachnoclostridium* in each group, (**F**): relative abundance of *Desulforibrio-piger* in each group). “*” is significantly different, and “**” is extremely significantly different.

**Table 1 animals-13-01447-t001:** Composition and nutrient composition of basal ration (air-dried level).

Items	Age (d)	
	1–28	29–56
Ingredients (air-dried basis, %)		
Corn	54.4	53
Soybean meal	23.6	16
Expanded soybean	5	3
Rice distiller’s grains	5	8
Rice bran	/	8
Corn bran	/	2
Soybean oil	2.2	4.5
Limestone	1.5	1.9
Fermented soybean meal	2.5	/
High grade corn gluten meal	2.0	/
Calcium hydrogen phosphate	2.0	1.8
NaCl	0.3	0.3
Premix feed	1.5	1.5
Total	100.00	100.00
Nutritional level		
Metabolic energy (kcal/kg)	2983	3090
Crude protein (%)	20.4	17.2
Lysine (%)	1.18	0.96
Methionine (%)	0.55	0.44
Methionine + cysteine (%)	0.90	0.74
Tryptophan (%)	0.22	0.20
Threonine (%)	0.88	0.78
Calcium (%)	0.86	0.73
Total phosphorus (%)	0.70	0.71
Nonphytate phosphorus (%)	0.43	0.44

The following substances are supplied per kilogram of diet: Vitamin A, 10,000 IU; Vitamin D_3_ 2500 IU; Vitamin E, 20 mg; Vitamin K, 3.5 mg; Vitamin B_1_, 1.5 mg; Vitamin B_2_, 3.5 mg; Vitamin B_6_, 3 mg; Vitamin B_12_, 10 μg; Pantothenic acid, 10 mg; Niacin, 30 mg; Biotin, 0.15 mg; Choline chloride, 1000 mg; Iron, 80 mg; Copper, 8 mg; Manganese, 60 mg; Zinc, 60 mg; Selenium, 0.15 mg; Iodine, 0.18 mg. “/” means that the substance is not added.

**Table 2 animals-13-01447-t002:** Effects of *Yucca saponin*, *Yucca schidigera*, and *Quillaja saponaria* on growth performance of broilers.

Items	Groups				SEM	*p*-Value
	CON	YSa	YS	QS		
Body weight, g						
1 d	33.7	32.4	33.3	31.6	1.49	0.188
28 d	455 ^d^	467 ^c^	480 ^b^	489 ^a^	14.33	<0.001
56 d	1555 ^b^	1599 ^a^	1633 ^a^	1624 ^a^	41.79	0.001
Average daily gain, g/d						
1–28 d	15.0 ^d^	15.6 ^c^	16.0 ^b^	16.4 ^a^	0.65	<0.001
29–56 d	39.3	40.5	41.2	40.7	1.46	0.062
1–56 d	27.2 ^b^	28.0 ^a^	28.6 ^a^	28.6 ^a^	0.97	0.006
Average daily feed intake, g/d						
1–28 d	32.0 ^c^	35.1 ^a^	34.1 ^b^	35.6 ^a^	1.95	<0.001
29–56 d	123	123	124	131	10.44	0.391
1–56 d	76.6 ^a^	78.4 ^ab^	78.5 ^ab^	82.4 ^b^	5.91	0.148
Feed: Gain, g: g						
1–28 d	2.17	2.22	2.09	2.12	0.10	0.086
29–56 d	3.20 ^a^	2.97 ^b^	2.91 ^b^	3.04 ^a^	0.20	0.044
1–56 d	2.87 ^a^	2.76 ^ab^	2.67 ^b^	2.76 ^ab^	0.13	0.065

^a–d^ Within a row, means with different superscripts differ significantly (*p* < 0.05). “CON” means broilers group fed with basic diet. “YSa” means broilers group fed with basic diet + 500 mg/kg YSa. “YS” means broilers group fed with basic diet + 500 mg/kg YS. “QS” means broilers group fed with basic diet + 500 mg/kg QS.

## Data Availability

The data presented in this study are available upon request from the corresponding author.

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
