# Peer review of "Comparison of the Effects of Yucca saponin, Yucca schidigera, and Quillaja saponaria on Growth Performance, Immunity, Antioxidant Capability, and Intestinal Flora in Broilers"

_animals, 2023, doi:10.3390/ani13091447_

Round 1
Reviewer 1 Report
This study is generally considered to be a good work.
However, there are questions about 'Materials and Methods'.
The cage size was described in this study, but I wonder if one cage is one replication.
If one cage is one repetition, it is likely that fewer broilers were housed for the cage size.
To see blood samples and microbes, I think we could have bred and tested them in larger numbers.
Another thing is that the breed of the broiler is not presented, so the analysis of the results is ambiguous.
The feed conversion rate (feed:gain) was high in the growth performance results, which seems to be higher than the previous broiler's growth performance.
These results are comparable to among the treatment groups, but it seems that they hard to compare to other studies.
Please review the material one more time.
Author Response
Dear Prof,
Please see the attachment about our responses.
Best regards,
Dai Zhenglie

Reviewer 2 Report
Very interesting and innovative work, with small considerations in the text just to improve understanding.
The discussion can be improved by inserting information about the composition of the three tested products, such as the amount of phenolic compounds present in each one, which may have contributed to the observed beneficial effects. Most of the effect was justified by the content of saponins, but other compounds present may also have contributed, but this information is missing in the text.

Author Response
Dear Prof,
Please see our responses in the attachment.
Best regards,
Dai Zhenglie
